# Use of a Taguchi Design in *Hibiscus sabdariffa* Extracts Encapsulated by Spray-Drying

**DOI:** 10.3390/foods9020128

**Published:** 2020-01-24

**Authors:** Migdalia Sarahy Navidad-Murrieta, Alejandro Pérez-Larios, Jorge Alberto Sanchéz-Burgos, Juan Arturo Ragazzo-Sánchez, Gabriel Luna-Bárcenas, Sonia G. Sáyago-Ayerdi

**Affiliations:** 1Tecnológico Nacional de México/Instituto Tecnológico de Tepic, Av. Tecnológico 2595, Lagos del Country, Tepic 63175, Nayarit, Mexico; murrieta_26@hotmail.com (M.S.N.-M.); jragazzo@ittepic.edu.mx (J.A.S.-B.); jsanchezb@ittepic.edu.mx (J.A.R.-S.); 2División de Ciencias Agropecuarias e Ingenierías, Centro Universitario de los Altos/Universidad de Guadalajara, Av. Rafael Casillas Aceves 1200, Tepatitlán de Morelos 47600, Jalisco, Mexico; alarios@cualtos.udg.mx; 3Centro de Investigación y Estudios Avanzados (CINVESTAV) del Instituto Politécnico Nacional, Unidad Querétaro, Libramiento Norponiente 2000, Fracc. Real de Juriquilla, Santiago de Querétaro 76230, Qro., Mexico; gabriel.luna@cinvestav.mx

**Keywords:** *Hibiscus sabdariffa*, phenolic compounds, spray-drying, antioxidant capacity

## Abstract

Aqueous and ethanolic extracts of *Hibiscus sabdariffa* were spray-dried using maltodextrin (MD) and gum arabic (GA) as carrier agents. A Taguchi L8 experimental design with seven variables was implemented. Physicochemical properties in the encapsulates were evaluated by Ultraviolet-Visible (UV-Vis,) X-ray Diffraction (XRD), spectroscopy and gravimetric techniques. Treatments with aqueous extracts showed the highest concentration of total soluble polyphenols (TSP) 32.12–21.23 mg equivalent gallic acid (EAG)/g dry weight (DW), and antioxidant capacity (AOX) in the 2,2-azinobis-3-ethylbenzotiazoline-6-sulfonic acid (ABTS) assay. The best treatment for TSP and AOX was T4: 2.5% *Hibiscus*
*w*/*w*, aqueous extract, decoction, extract-to-carrier ratio 1:1 (*w*/*w*), proportion to carriers (MD:GA) 80:20 (*w*/*w*), 10,000 rpm, 150 °C. The Taguchi L8 design is a tool that allows the use of multiple variables with a low number of treatments that indicate the drying conditions that give the best parameters, focusing mainly on TSP and AOX, also, it is a good alternative for the preservation and stability of the phenolic compoudns in *Hibiscus*.

## 1. Introduction

*Hibiscus sabdariffa* L. is a shrub that belongs to the Malvaceae family, is a good source of dietary fiber (DF) rich in non-starch polysaccharides, and phenolic compounds (PC) such as flavonoids (kaempferol, quercetin, hibiscetin), anthocyanins (cyanidin, delphinidin. hibiscin), and phenolic acids (protocatechuic acid, chlorogenic acid, hibisicus acid) [1,2]. Some enhanced varieties like ‘Cruza Negra’ have a high concentration of PC (41.52 mg mg equivalent gallic acid (EAG)/g dry weight (DW)) and high antioxidant capacity (AOX) [3]. PC extracts from *Hibiscus* can be used in diverse functional applications such as antioxidant, antihypertensive, effectiveness against low-density lipoprotein oxidation, and hyperlipidemia [2], but they are sensitive to UV rays, oxygen, and high temperatures [4]. Spray-drying encapsulation has been demonstrated to be a useful and economical technique to preserve chemical stability, and PC structure [4]. The selection of a carrier agent depends upon the physicochemical properties and the final application of the product. Maltodextrin (MD) is one of the most common carrier agents used for encapsulation that shows low viscosity at high solid content, and high solubility like to gum Arabic (GA) [5]; both carriers are the preferred choice because of their biocompatibility and innocuous nature [5]. This film creates a net with hydrophobic and/or hydrophilic properties that protects the PC from external agents [6]. Encapsulation of PC from *Hibiscus* by spray drying has been previously studied but focusing in the effect of spray-drying temperature in *Hibiscus* extracts and the effect on the volatile compound loss; in the evaluation different type of carriers’ on the release kinetics of *Hibiscus* extract, and using modified achira (*Canna indica L.*) starch as carrier agent for the encapsulation of the *Hibiscus* extract [7,8,9]. In this regard, it is important to determine the best conditions of multiple variables to optimize the drying conditions. The use of an experimental Taguchi design allows evaluation of the multiple variables that influence spray-drying. The design provides the best values of variables to ensure the quality of the product/process with few treatments [10].

The use of an experimental Taguchi design allows one to evaluate multiple variables that influence spray-drying at different levels with a small number of experiments. The Taguchi experimental design reduces cost, improves quality and provides robust design solutions. This method has evolved into an established approach for analyzing interaction effects when ranking and screening various factors [11]. Since it reduces the number of experiments, it is easy to use, accurate and reliable, saving time and cost. Taguchi designs can determine the experimental conditions having the best effect on the desired characteristics, usually fixed by the investigator [12,13]. The analysis of variance (ANOVA), range analysis and analysis of signal-to-noise ratio (S/N) are the main analysis methods for this design [13]. The variability is expressed by signal to noise (S/N) ratio, the maximum S/N ratio is considered as the optimal condition as the variability characteristics is inversely proportional to the S/N ratio [14], but the appropriate result will mostly depend on the selection of the most influential factors and their levels [13].

Taguchi methodology has applications in different food, biological and/or biotechnological areas; these include spray-drying, fermentations, medicine, pharmacy, food processing, among others [11,12,13,14,15,16,17,18].

This study aimed to evaluate the effect of spray-drying variables, in an aqueous and ethanolic extract of *Hibiscus sabdariffa* on the PC content and AOX by the experimental Taguchi L8 design, as well as analyze the physicochemical properties of the encapsulated products. Seven variables (extraction solvent, *Hibiscus* concentration, decoction, extract-carriers’ ratio, the proportion of carriers’ agents, homogenization and dried inlet temperature) with two levels and mixtures of MD and GA utilized as carrier agents.

## 2. Materials and Methods

### 2.1. Materials

*H. sabdariffa* “Cruza Negra” calyces were provided by a local producer (Tepic, Nayarit, México, 21° 39′ 15′′ N, 106° 32′ 45′′ O). The samples were ground (500 µm) and mixed to obtain homogenous samples. They were preserved in hermetic metallic bags. The carriers’ used were corn maltodextrin (MD; DE 10) (Tate & Lyle Ingredients, Decatur, IL, USA), and gum arabic (GA) (Nexira Food, Rouen, France).

### 2.2. Experimental Design

A Taguchi L8 design was performed to obtain the best extraction conditions base upon PC and AOX contents. Seven variables with two levels were used (Table 1). 

The experimental design was composed of eight treatments of the independent variables. All experiments were performed in triplicate. The levels were selected on basis reported studies [19,20,21,22,23]. To avoid systematic errors, all the experiments were performed in a random order, to minimize the effect of unexplained variability on the responses obtained.

### 2.3. Sample Preparation and Spray-Drying of Hibiscus Extracts

The *Hibiscus* extracts were prepared according to the conditions indicated in Table 1; the extracts were stored in amber flasks at 4 ± 2 °C. The solid content of the extracts and the carriers’ materials (MD and GA) were dispersed in water and homogenized (Ultra-Turrax T18 with disperser S18N-19G; IKA, Staufen, Germany) (Table 1) during 5 min. The prepared carriers’ agents’ solutions were combined with *Hibiscus* extract and homogenized again at the same conditions. The feed mixtures were spray-drying (B-290, Büchi Labortechnik AG, Flawil, Switzerland) with a main chamber of 165 mm diameter, 600 mm cylindrical height and 1.5 mm nozzle diameter. The pump power was kept at 15% to maintain feed flow rate as 4.5 mL/min, the airflow rate was 38 m^3^/h and 3 pulse clean for a period. During drying processes, the temperature of the feed mixture was 24°C at constant magnetic stir (Cimarec digital, Thermo Scientific, Waltham, MA, USA). The microencapsulates obtained were evaluated in their physicochemical properties, total soluble polyphenols (TSP) and AOX capacity to obtain the best extraction condition.

### 2.4. Physicochemical Properties

#### 2.4.1. Encapsulation Yield (EY)

EY was evaluated by the material balance of the product recovery given by the perceptual ratio (dry basis) between the total mass of products recovered by the mass of juice fed to the system [23].

#### 2.4.2. Encapsulation Efficiency (EE)

The EE is the percentage ratio of encapsulated PC content and total PC content (Eq. 1), it was determined to take the difference of total PC (TPC) and the PC at time 0 when is reconstituting the powder in the solvent which is called surface PC (SPC) following the methodology described in the Section 2.8. for total soluble polyphenols (TSP) [24]:(1)$$\mathrm{EE}\;(\%)\;=\;\frac{\left(\mathrm{TPC}-\mathrm{SPC}\right)}{\mathrm{TPC}}\;\times 100$$

#### 2.4.3. Moisture, Aater Activity (Aw), pH, Solubility, Wettability and, Bulk Density Analysis

Partially air-dried samples were kept in a hot-air oven at 110 °C until they reached a constant weight, and the moisture content was calculated in terms of the weight loss method 925.10 [25] (AOAC, 2012). An Aw meter (Aqualab 4TEV, Decagon Devices, Pullman, WA, USA) was used. Five g of sample was placed inside the chamber where the Aw was determined by the dew point principle; where the water activity is the relative humidity of air (vapor phase water) in equilibrium with the liquid phase water of the sample, in a sealed chamber. For pH, dilutions 1:100 (*w*/*v*) of the encapsulates were prepared in distilled water and the pH was determined by a potentiometer (Hanna HI 2210, Woonsocket, RI, USA) at 25 °C [26]. The solubility in encapsulates was evaluated according to an existing method [27]. The wettability was evaluated using an existing static method [28], this one was expressed as the time necessary for 1 g of encapsulates to disappear from the water surface. The bulk density was evaluated according to a reported method [29].

### 2.5. Scanning Electron Microscopy (SEM)

The morphology of the spray-dried particles was visualized using scanning electron microscopy (SEM) (Tescan, MIRA3 LMU, London, UK), at 5 kV with a magnification of 5000×. Samples were mounted on self-adhesive tape and gold-coated before imaging (Denton Vacuum Desk V operated at 10 mA for 60 s, Moorestown, NJ, USA)

### 2.6. Absorption Spectrum

An UV-Visible spectra 200–900 nm was used to evaluate the absorption spectrum in the treatments, MD and GA alone, and the mix of carriers. The distribution was carried out by Dynamic Light Scattering (DLS) using a UV 2600 spectrophotometer (Shimadzu, Kyoto, Japan).

### 2.7. X-ray Diffraction (XRD)

Encapsulates were analyzed in a diffractometer (Bruker D8, Tokyo, Japan) (Kα Cu =1.5460 Å, 40 kV, 30 mA), considering the diffraction intensity as a function of the diffraction angle (2θ) between 10° and 90°, using a step of 0.02° and 0.25 s per step.

### 2.8. Total Soluble Polyphenols (TSP) Content and AOX Assays.

The encapsulates were extracted to evaluate the TSP according to an existing method [30]. TSP contents were determined in the supernatants using an existing method [31] in a microplate reader (BioTek, Synergy HT, Winooski, VT, USA). The absorbance was read at 750 nm against a blank, and TSP was calculated using the calibration curve of gallic acid. The results were expressed as gallic acid equivalents (mg GAE/g dry weight (DW)).

The supernatants were used to evaluate the DPPH radical scavenging method, 2,2-azinobis-3-ethylbenzotiazoline-6-sulfonic acid (ABTS) analysis and, ferric reducing antioxidant power (FRAP) assay. For DPPH assay was analyzed by modifying an existing method [32,33], the absorbance was read at 517 nm, the results are reported in mmol TE (Trolox equivalent)/g DW. For the ABTS radical assay based in an existing method [34] with some modifications [33]; the absorbance was read at 734 nm, the results were reported in mmol TE/g DW. For FRAP assay was analyzed by an existing method [35]; Trolox was used as a standard and methanol acidified as blank, the absorbance was read at 595 nm and the results were reported in mmol TE/g DW.

### 2.9. Identification of PC by HPLC-DAD

Partial identification of PC was done according to the proposed method using an Agilent 1260 series high pressure liquid chromatography (HPLC) system (Agilent Technologies, Santa Clara, CA, USA) equipped with a UV–Vis diode array detector (DAD) [36]. The chromatographic separation was performed in a Poroshell 120 EC-C18 column (4.6 mm × 150 mm, particle size 2.7 μm) (Agilent Technologies) at a flow rate of 0.4 mL/min using an injection volume of 10 μL. The mobile phases consisted in water containing formic acid (0.1 cm^3^/100 cm^3^) as solvent A and acetonitrile as solvent B, following multi-step linear gradient was applied: 0 min, 5% B; 10 min, 23% B; 15 min, 50% B; 20 min, 50% B; 23 min, 100% B; 25 min, 100% B; 27 min, 5% B; 30 min, 5% B; finally, the initial conditions were held for one min to equilibrate the system before the subsequent injection. The PCs were monitored in a range of 280–320 nm for the DAD.

### 2.10. Data Analysis

The data were expressed as mean ± standard deviation (SD) for each treatment. The *p*-values were used as a tool to check the significance of the effects of the variables considered: Hibiscus concentration, solvent of extraction, decoction, extract: carriers ratio, carriers’ ratio (MD + GA), homogenization, inlet temperature (Table 1) in the response variables: TSP content and AOX (Appendix A), through an ANOVA using the STATISTICA software, version 10.0 (StatSoft Inc. 1984–2007, Tulsa, OK, USA).

The S/N ratio analysis was realized for each level of process parameters. The category of the quality was the-larger-the better where the optimal level of the process parameters was the level with the greatest S/N ratio ETA= −10 × log10(1/N × Sum (1/y2)) (Appendix A). This was the base for the decision of the optimum level for each factor. A parameter effects plot was then generated from the results of the analysis of means.

For each parameter, an ANOVA analysis was performed using a univariate design to determine the differences between the treatments. For means comparison, a Fisher LSD test (Fisher’s least significant difference test) with a significance level of α = 0.05 was applied.

## 3. Results and Discussion

### 3.1. Physicochemical Properties

The EY values obtained were from 61.88% to 89.39% in T2 (2.5% *Hibiscus*, water, decoction, extract: carriers’ 1:2, ratio carriers’ 90:10, 5000 rpm, 110 °C) treatment (Table 2). High inlet temperatures cause more proximity of particles to their glass transition temperatures, and more adherence, especially if the droplet still has a high moisture content [37]. The extract/carrier ratio is another factor to consider, as treatments with a 1:2 (*w*/*w*) ratio have a higher carrier agent content and therefore higher temperature tolerance resulting in higher EY [27]. Another investigation showed similar results in spray-drying with 89 ± 3% using MD and GA [28]. The EE ranged from 80.94 to 88.48% in T5 (1% *Hibiscus*, water, no decoction, extract:carrier 1:2, carrier ratio 90:10, 10,000 rpm, 150 °C). The relationship between type of carrier, core/coating ratio, ultrasonication time, coating material types, content of PC, and particle size influence the EE [6,9,24]. Reports indicate that the core (material to be encapsulated) shows physicochemical properties that must be taken into account during the encapsulation process [38]. The decrease in the amount of TSP is attributed to the high inlet temperature, causing degradation by structural chemical changes of the PC [9], other factors are the extract:carrier ratio, and the extraction conditions (decoction and weight of *Hibiscus*) [5].

The physicochemical properties (Table 2) showed the highest moisture value was obtained in T3 (1% *Hibiscus*, water, no decoction, extract:carriers 1:1, carrier ratio 80:20, 5000 rpm, 110 °C) with 9.55% ± 0.41 and the lowest, obtained in four treatments—T1, T6, T7 and T8 (all with ethanolic extraction)—ranged between 2.77–4.89%; this result can be attributed to the concentration of total soluble solids at the beginning of the spray-drying that generates a decrease in this parameter [37] (Table 2). The decrease in moisture content is a function of the drying temperature; under high temperature, the rate of heat transfer within the particle is high, which leads to a faster evaporation of water, thereby the moisture content of the product decreases [5].

The Aw showed significant differences (*p* < 0.05) in all the treatments. Aw contents obtained were low, this parameter provides the guideline to predict the stability and shelf life in a product [38]. The values obtained in the pH parameter showed significant differences (*p* < 0.05), the acidity can be due to organic acids from *Hibiscus* such as malic, citric, hibiscus, and hydroxycitric acids (Figure 4) [1]. The solubility (%) did not show significant differences, this parameter is related to the moisture of the particles [9]. Other factor that influences the solubility is the proportion of the carriers’ (MD and GA). MD and GA are widely used in the process of spray-drying due to high solubility in water, and for its emulsification properties [39].

In the wettability parameter, the highest value was obtained for T7 (2.5% *Hibiscus*, ethanol 20%, no decoction, extract:carriers 1:2, carriers ratio 80:20, 10000 rpm, 110 °C) with 7.72 min ± 0.42. Reports indicate that the wettability process corresponds to the phenomenon that allows the penetration of water in a particle [28]. According to reports, wetting times of 2.38–1.62 min, have been expressed for a *Hibiscus* extract in spray-drying powder without drying agents [26]. The wettability is directly affected by the molecular interactions between the components [40]. Authors suggest that a higher moisture content is probably the generation of a greater number of hydrophilic groups on the particles, thus reducing time through greater interaction with water [39]. In bulk density, the highest value was T4 (2.5% *Hibiscus*, water, decoction, extract:carriers 1:1, carrier ratio 80:20, 10,000 rpm, 150 C) with 0.46 g/cm^3^ ± 0.03; these results were similar to those reported by others authors but lower than those reported for *Hibiscus* extracts without carrier agents (0.73–0.87 g/cm^3^) [26,28]. The bulk density of the encapsulates is related to the molecular weight of the carrier agents in a heavier material it can fit more easily in the spaces between the particles, occupying less space and giving higher apparent density values [41].

### 3.2. Scanning Electron Microscopy (SEM)

In Figure 1 is shown an SEM of one encapsulate of *Hibiscus* extract obtained by spray-drying. It shows spherical particles of different diameters; as well as slightly smooth surfaces attributable to the carrier agents (MD and GA) with the formation of some concavities with a wrinkled surface. The formation of concavities on the surface of the atomized particles can contribute to a contraction during drying [42]. The formation of wrinkles occurs because of the slow formation of a cover film when droplets are atomized, mainly influenced by drying temperature (see Appendix A) [34].

### 3.3. Absorption Spectrume

Figure 2 shows the carriers’ agents spectrum in the mixture and individually, in which it is possible to identify two absorption bands at 211 and 282 nm (Figure 2a). These bands are characteristic of carbonyl groups derived from the conformational structures of MD and GA. 

Figure 2b shows all the treatments obtained, where it is possible to identify a third weak band at 327 nm. According to reports, this band is attributed to the presence of PC such as catechins, flavonoids and phenolic acids [43]; these phytoconstituents are characterized by the presence of various chromophores and conjugated systems that act as UV-absorbing systems. The presence of anthocyanins from *Hibiscus* was evidenced in the band at 532 nm [1]. These spectra indicate a homogeneous behavior between treatments, as well as, a rearrangement between the PC of *Hibiscus* and the carrier agents showing specific bands for each of the signals indicating interactions attributable to hydrogen bonding [44].

### 3.4. X-ray Diffraction

The X-ray diffraction profiles (XRD), indicate that the MD presents a crystalline structure with nine peaks at 2theta scale = 10°, 11.3°, 13.1°, 14.1°, 17°, 21.2°, 26.6°, 28.6° and 30.1° (Figure 3), the GA also presents a characteristic peak at 18°, on the 2theta scale, which suggests that the GA and the MD are encapsulated with the different treatments, the intensity of the peaks decreases, which generate molecular interactions between the carriers’ agents and PC that structurally modify the diffraction profile, resulting in low intensity of the signal and excessive noise, this can describe the characteristic presence of amorphous zones [38]. Similar results were reported indicating the presence of amorphous material when mango juice was dehydrated by spray-drying using different carriers’ agents [27]. Therefore, according to the XRD patterns in the different treatments, there is a diversity of compounds present in the samples that generate molecular interactions between the carrier’s agents that structurally modify the diffraction profile [38].

### 3.5. PC Content and AOX in Hibiscus Encapsulates

The TSP and AOX in encapsulates of *Hibiscus* extracts (Table 3) showed significant differences between treatments (*p* < 0.05). The highest content in TSP was for T4 (2.5% *Hibiscus*, water, decoction, extract: carrier 1:1, carrier ratio 80:20, 10,000 rpm, 150 °C) with a concentration of 32.13 ± 0.06 mg EAG/g DW; and the lowest to T6 (1% *Hibiscus*, ethanol 20%, decoction, extract:carrier 1:2, carrier ratio 80:20, 5000 rpm, 150 °C) treatment with a 7.16 ± 0.05 mg EAG/g DW, this concentration of TSP can be explained due to the extraction solvent (EtOH 20%), and the extract:carrier ratio (1:2) (see Appendix A). TSP values reported for the same ‘Cruza Negra’ variety under the same extraction method were similar (41.52 ± 0.99 mg EAG/g DW) [3]. The polar solvents’ influence the extraction because the major PC in *Hibiscus* are flavonoids with glyosidic structures and are more soluble (see Appendix A) [45]. For that the extraction of gallic and chlorogenic acid and (+)–catechin present in *Hibiscus* could be considered (Figure 4) [46].

The AOX in the aqueous extracts showed two-fold higher values than ethanolic extracts with a correlation with the results of TSP (see Appendix A). When the polarity of the solvent increases (see Appendix A) are higher EY of total soluble solids and extractable PC [47].

For the evaluation of antiradical activity (ABTS), the treatment with the highest value corresponds to T4 (2.5% *Hibiscus*, water, decoction, extract: carrier 1:1, carrier ratio 80:20, 10,000 rpm, 150 °C) 216.28 ± 5.24 mmol ET/g DW (Table 3); similar values were reported with values from 221.5–363.2 mmol ET/g DW for different varieties of *Hibiscus* [45]. This can be explained by the heat treatment to which the calyces are subjected for extraction (see Appendix A), and the hydrolysis generated by the acidity gave that the ellagic and gallotannin tannins (Figure 4) [46].

The activity for the DPPH radical, attributed to the solvent of extraction (water) and the extract: carrier ratio (1:1); when the encapsulate containing a smaller amount of carrier agents there are more hydroxyl groups available to react with the DPPH radical [43]. As well, this can be influenced by the condensed tannins derived from the heat treatment of calyces, since they exhibit a good (DPPH) AOX capacity mainly due to the tannins and flavonoids containing a variety of hydroxyl groups that show stronger AOX and elimination of free radicals of the PC present (see Appendix A) [48]. The chelating activity (iron reducing antioxidant power—FRAP), T3 (1% *Hibiscus*, water, no decoction, extract:carrier 1:1, carrier ratio 80:20, 5000 rpm, 110 °C) showed highest AOX with 271.95 ± 6.63 mmol ET/g DW value; according to the results (see Appendix A), all variables influence significantly this AOX (FRAP), within the main ones are the extraction solvent, extract:carrier agents ratio, carrier ratio (MD + GA), homogenization and inlet temperature. The homogenization is an important point to take into consideration, because at high-energy homogenization the particle size will be smaller, and a lower temperature will be required in the drying process and, the content and proportion of the carriers’ agents will be smaller (see Appendix A) [49]. Previous in vitro studies have demonstrated the ability of flavonoids such as catechins and quercetin to inhibit the formation of free radicals and chelation of metal ions, particularly those of iron and copper [50]. Higher AOX values were presented by the FRAP and ABTS techniques, can be attributed to the presence of gallic and chlorogenic acid (Figure 4), as well as, the presence of phenolic acids, flavonoids, and anthocyanins present in *Hibiscus* (p.e. catechin and quercetin) which, given the position of the hydroxyl groups, are responsible for the high chelating activity and the radical quenching power [46].

## 4. Conclusions

The results the Taguchi L8 design allowed us to obtain the best values in the studied variables to ensure the quality of the *Hibiscus sabdariffa* extracts encapsulates. The T4 treatment showed the highest concentration of TSP with AOX. A good EY and EE (%) and storage stability showed the interactions between the carrier agents and the PC present in the calyces. These new interactions were confirmed by the results of UV-Vis spectroscopy and X-ray diffraction. The use of an experimental Taguchi design is a useful methodology that uses a low number of experiments to evaluate the interactions between parameters; this indicates that the maximum TSP content and AOX in the *Hibiscus sabdariffa* extracts encapsulates were mainly related to the solvent of extraction, extract: carrier agent ratio, and the carrier agents ratio (MD + GA).

## Figures and Tables

**Figure 1 foods-09-00128-f001:**
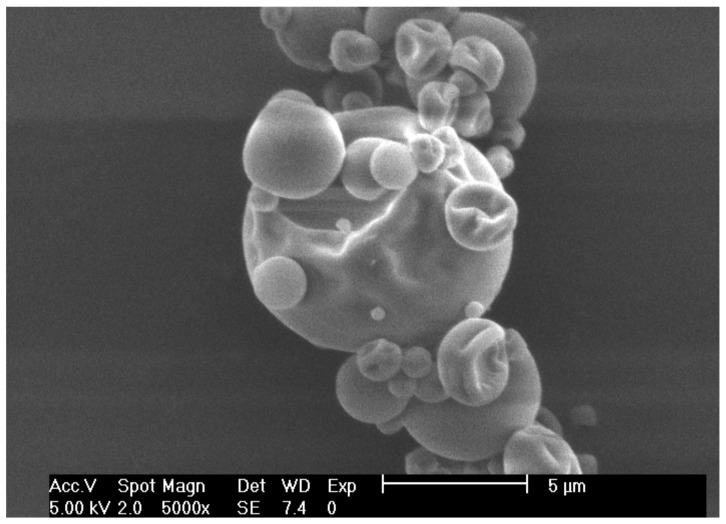
SEM micrograph of *Hibiscus sabdariffa* extract encapsulated by spray-drying.

**Figure 2 foods-09-00128-f002:**
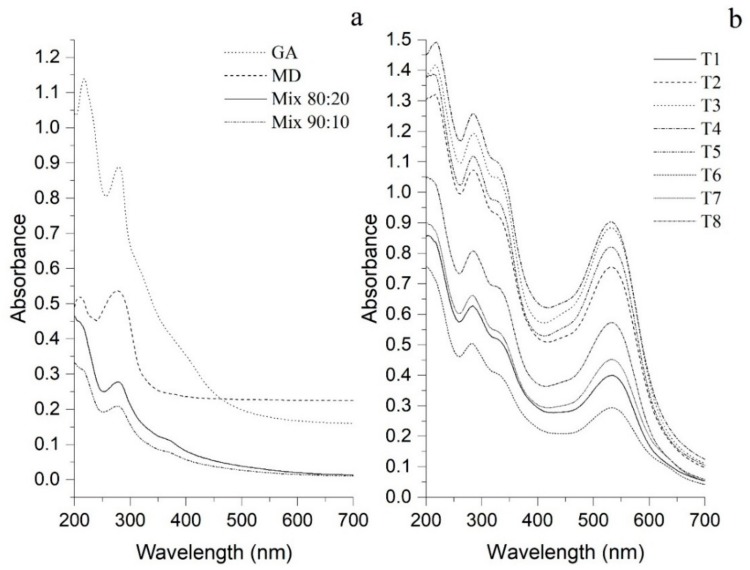
Absorption spectrum. (**a**) Only carriers’ and 80:20 and 90:10 mixtures of carriers (MD DE. 10:GA). (**b**) Treatment extracts of *Hibiscus (Hibiscus sabdariffa*) spray-drying.

**Figure 3 foods-09-00128-f003:**
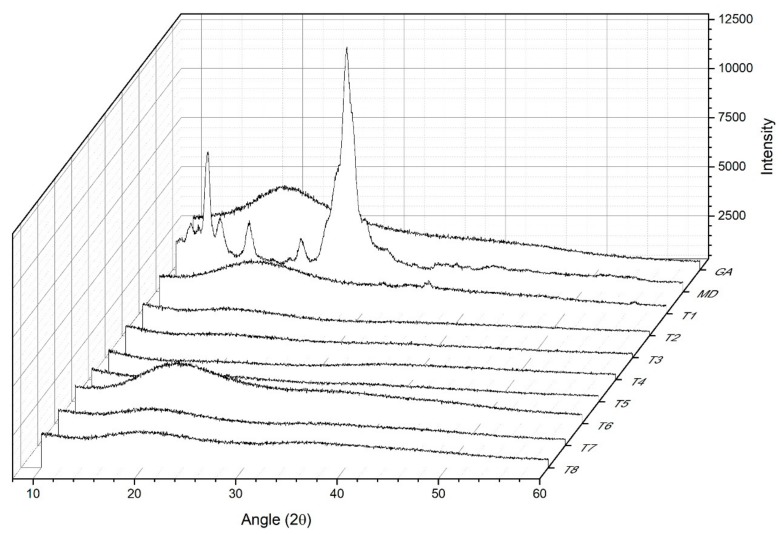
X-ray diffraction for treatment extracts of *Hibiscus (Hibiscus sabdariffa*) spray-drying. MD: maltodextrin DE 10 and GA: gum Arabic.

**Figure 4 foods-09-00128-f004:**
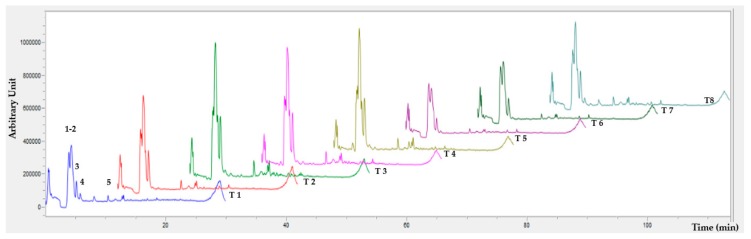
PC profile analyzed by HPLC-DAD of extracts of *Hibiscus (Hibiscus sabdariffa*) spray-drying. (1) Hibiscus acid hydroxyethyl ester, (2) Hydroxycitric acid, (3) Hibiscus acid, (4) Gallic acid and (5) Chlorogenic acid.

**Table 1 foods-09-00128-t001:** Experimental design Taguchi L8 of treatments of extracts of *Hibiscus (Hibiscus sabdariffa)* spray-drying ^1^.

	*Hibiscus* Concentration (%)	Solvent of Extraction	Decoction (°C/min)	Extract: Carrier Ratio (*w*/*w*)	Carriers Ratio (MD + GA) ^2^ (%)	Homogenization (rpm)	Inlet Temperature (°C)
**T1**	1	Ethanol 20%	100/5	1:1	90:10	10,000	110
**T2**	2.5	Water	100/5	1:2	90:10	5000	110
**T3**	1	Water	NA	1:1	80:20	5000	110
**T4**	2.5	Water	100/5	1:1	80:20	10,000	150
**T5**	1	Water	NA	1:2	90:10	10,000	150
**T6**	1	Ethanol 20%	100/5	1:2	80:20	5000	150
**T7**	2.5	Ethanol 20%	NA	1:2	80:20	10,000	110
**T8**	2.5	Ethanol 20%	NA	1:1	90:10	5000	150

^1^ Values are the mean ± standard deviation (*n* = 3). Different letters in each column indicate significant a difference. Mean analyzed by LSD (*p* < 0.05). NA: Not apply. ^2^ MD: Maltodextrin DE 10, GA: Gum arabic.

**Table 2 foods-09-00128-t002:** Physicochemical analysis to treatments of *Hibiscus* extracts (*Hibiscus sabdariffa*) spray-drying with different proportions of MD and GA ^1^.

Treatment	EY ^2^ (%)	EE ^3^ (%)	Moisture (%)	A_w_	pH	Solubility (%)	Wettability (min)	Bulk Density (g/cm^3^)
T_1_	61.88 ± 0.93 ^b^	86.58± 0.95 ^bc^	4.89 ± 0.87 ^abc^	0.27 ± 0 ^b^	3.09 ± 0.01 ^e^	89.22 ± 2.0 ^a^	5.14 ± 0.03 ^ab^	0.35 ± 0.02 ^f^
T_2_	89.38 ± 1.04 ^g^	85.45± 4.47 ^bc^	5.97 ± 0.86 ^c^	0.23 ± 0 ^a^	2.75 ± 0.01 ^c^	77.11 ± 5.97 ^a^	4.76 ± 0.42 ^ab^	0.44 ± 0.01 ^cde^
T_3_	73.53 ± 1.01 ^a^	87.93± 2.94 ^c^	9.55 ± 0.41 ^f^	0.38 ± 0 ^h^	2.70 ± 0.01 ^b^	83.82 ± 3.24 ^a^	2.95 ± 0.16 ^a^	0.43 ± 0.03 ^bcd^
T_4_	86.34 ± 0.05 ^f^	86.70± 2.23 ^bc^	5.19 ± 0.62 ^bc^	0.35 ± 0 ^e^	2.65 ± 0.01 ^a^	86.06 ±0.33 ^a^	3.08 ± 1.51 ^a^	0.48 ± 0.03 ^e^
T_5_	69.01 ± 0.42 ^c^	88.48± 0.73 ^c^	8.06 ± 0.66 ^e^	0.36 ± 0 ^f^	2.65 ± 0.01 ^a^	82.23 ± 3.05 ^a^	3.92 ± 0.46 ^a^	0.46 ± 0.03 ^de^
T_6_	74.80 ± 0.68 ^d^	82.61± 0.18 ^ab^	4.49 ± 0.58 ^ab^	0.29 ± 0 ^d^	3.55 ± 0.01 ^g^	90.82 ± 1.42 ^a^	6.19 ± 2.97 ^ab^	0.42 ± 0.03 ^abc^
T_7_	84.40 ± 0.28 ^e^	80.95± 1.13 ^a^	3.76 ± 0.35 ^ad^	0.28 ± 0 ^c^	3.19 ± 0.01 ^f^	89.71 ± 2.01 ^a^	7.72 ± 0.42 ^b^	0.40 ± 0.01 ^ab^
T_8_	72.53 ± 0.59 ^a^	85.36± 2.02 ^abc^	2.77 ± 0.84 ^d^	0.37 ± 0 ^g^	2.80 ± 0.01 ^d^	90.76 ± 0.76 ^a^	4.83 ± 2.23 ^ab^	0.38 ± 0.02 ^af^

^1^ Values are the mean ± standard deviation (*n* = 3). Different letters in each column indicate a significant difference. Mean analyzed by LSD (*p* < 0.05). ^2^ EY: Encapsulation yield. ^3^ EE: Encapsulation efficiency, NA, not apply. T_1_: 1% *Hibiscus*, ethanol 20%, decoction, extract:carrier 1:1, carrier ratio 90:10, 10,000 rpm, 110 °C.

**Table 3 foods-09-00128-t003:** Total soluble polyphenols (TSP) and antioxidant capacity (AOX) in treatments of extracts of *Hibiscus* (*Hibiscus sabdariffa*) spray-drying ^1.^

Treatment	TSP	AOX		
	(mg GAE/g DW) ^2^	(mmol TE/g DW) ^3^		
		ABTS	DPPH	FRAP
T_1_	16.00 ± 0.56 ^d^	103.16 ± 3.79 ^b^	28.60 ± 3.30 ^ab^	220.06 ± 2.27 ^a^
T_2_	21.23 ± 0.17 ^e^	139.64 ± 12.13 ^c^	57.61 ± 5.19 ^e^	258.43 ± 1.13 ^f^
T_3_	30.51 ± 0.41 ^g^	201.80 ± 0.05 ^e^	89.15 ± 11.00 ^d^	271.95 ± 6.63 ^g^
T_4_	32.13 ± 0.06 ^h^	216.28 ± 5.24 ^f^	87.78 ± 2.76 ^cd^	231.41 ± 0.13 ^e^
T_5_	25.69 ± 0.11 ^f^	155.16 ± 0.12 ^d^	80.12 ± 4.78 ^c^	224.45 ±1.29 ^a^
T_6_	7.16 ± 0.05 ^a^	24.27 ± 0.01 ^a^	21.29 ± 0.02 ^b^	67.97 ± 1.29 ^b^
T_7_	9.14 ± 0.03 ^b^	24.42 ± 0.11 ^a^	30.67 ± 0.42 ^a^	140.56 ± 0.83 ^c^
T_8_	13.73 ± 0.21 ^c^	24.04 ± 0.04 ^a^	33.02 ± 0.04 ^a^	154.09 ± 0.20 ^d^

^1^ Values are the mean ± standard deviation (*n* = 3). Different letters in each column indicate a significant difference. Mean analyzed by LSD (*p* < 0.05). ^2^ GAE, gallic acid equivalent, dry weight (DW). ^3^ TE, Trolox equivalent, dry weight (DW).

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
