# Peer review of "Use of a Taguchi Design in Hibiscus sabdariffa Extracts Encapsulated by Spray-Drying"

_foods, 2020, doi:10.3390/foods9020128_

Round 1
Reviewer 1 Report
The manuscript contains very important issues. The described parameters show a deep knowledge of the authors regarding the research topic. There are all typical parts of manuscript; introduction, methodology, results and discussion, conclusions. References contain 36 sources, are current and relevant to the topic of the manuscript. The language is correct.
The authors have done a lot of research that can have scientific and practical significance. They largely discussed them with available literature.
However regarding methodology it needs improvement. It contains too little information about Taguchi L8 design.
Taguchi L8 design is not often used in food research. No detailed information on data analysis. I am afraid that the analysis may be unreliable, because there are 7 factors and only 2 levels, and the experimental design was composed by only eight treatments of the independent variables. I am not sure have Authors used the right experimental design. It is also difficult to assess the interaction of various factors.
The use of small number of experiments is beneficial, but together with the repetitions 24 experiments have been performed! Other methods of planning the experiments reduce their number by using the repetitions only at a focal point.
The Authors used parameters coding, but still this manuscript is difficult to read because they often write all their values. As part of the description of the methodology, it is worth preparing one table with the plan of experiments and coding so as not to repeat it in the figures and tables..
Some others comments.
Line 69: “data not shown” - Is the data proprietary? if published, it is a good idea to provide a source.
Line 82: The inlet air temperature is given in table 1. What was the temperature of the outlet air in the dryer? This information should also be provided here.
Line 115: What was the examination about, did the mistake creep in?
Line 160: Which factor was analysed under "Analysis of variance (ANOVA) using a univariate design was performed to determine the differences between the treatments", if there were 7 factors?
Line 167: Why are the results of EY and EE presented in table placed in the methodology?
Line 263: The sentence to check
Line 341-347: Conclusion
I do not find the Taguchi L8 model used in this manuscript, no mention of it in the discussion of the results, I do not know what model the authors write about, because there is no information about using it to interpret and discuss the results obtained, no model description, no charts, no mathematical analysis, statistical .. It was only used to plan the experiment and what next?
Author Response
We have addressed all comments/revisions by the reviewers; the modified sections are highlighted in red text. We believe the revised manuscript has been greatly improved by taking into account all reviewers.
Please feel free to contact us if you require any further information.
Sincerely,
Sonia G Sayago Ayerdi
-----------------------------------------------------------------------------------------------------------------
The manuscript contains very important issues. The described parameters show a deep knowledge of the authors regarding the research topic. There are all typical parts of manuscript; introduction, methodology, results and discussion, conclusions. References contain 36 sources, are current and relevant to the topic of the manuscript. The language is correct.
The authors have done a lot of research that can have scientific and practical significance. They largely discussed them with available literature.
However regarding methodology it needs improvement. It contains too little information about Taguchi L8 design.
We have amended the Introduction as follows:
Lines 51-65:
“The use of an experimental Taguchi design allows one evaluating multiple variables that influence spray-drying at different levels with a low number of experiments. The Taguchi experimental design reduces cost, improves quality and provides robust design solutions. This method has evolved into an established approach for analyzing interaction effects when ranking and screening various factors [11]. Since it reduces the number of experiments, it is easy to use, accurate and reliable, saving time and cost. Taguchi design can determine the experimental conditions having the best effect on the desired characteristics usually fixed by the investigator [12,13]. The analysis of variance (ANOVA), range analysis and analysis of signal-to-noise ratio (S/N) are the main analysis methods for this design [13]. The variability is expressed by signal to noise (S/N) ratio, the maximum S/N ratio is considered as the optimal condition as the variability characteristics is inversely proportional to the S/N ratio [14]; but the appropriate result will mostly depend on the selection of the most influential factors and their levels [13].
Taguchi methodology have applications in different food, biological and/or biotechnological areas; these include spray drying, fermentations, medicine, pharmacy, food processing, among other [15, 11, 12, 13, 14, 15, 16, 17, 18].”
We have added the following text (Materials and Methods):
In line 82:
“A Taguchi L8 design was performed to obtain the best extraction conditions base upon PC and AOX contents.”
In line 84-85:
“All experiments were performed by triplicate”
In line 85:
References are added of relevant reported studies and deleted word “before”
In lines 188-201, the following text was amended:
“The data were expressed as mean ± standard deviation (SD) for each treatment. The p-values were used as a tool to check the significance of the effects of the variables considered: Hibiscus concentration, solvent of extraction, decoction, extract:carriers ratio, carriers ratio (MD + GA), homogenization, inlet temperature (Table 1) in the response variables: TSP content and AOX (Supplementary Materials S.2. and S.3.), through an ANOVA using the STATISTICA software, version 10.0 (StatSoft Inc. 1984–2007, Tulsa, EE. UU.).
The S/N ratio analysis was analyzed for each level of process parameters. The category of the quality was the-larger-the better where the optimal level of the process parameters is the level with the greatest S/N ratio ETA= -10*log10(1/N*Sum (1/y2))) (Supplementary Material S.3.). This is the foundation for the decision of the optimum level for each factor. A parameter effects plot was then generated from the results of the analysis of means.
For each of the parameters, an analysis was performed ANOVA using a univariate design was performed to determine the differences between the treatments. For means comparison, a Fisher LSD test (Fisher’s least significant difference test) with a significance level of α = 0.05 was applied”
Taguchi L8 design is not often used in food research. No detailed information on data analysis. I am afraid that the analysis may be unreliable, because there are 7 factors and only 2 levels, and the experimental design was composed by only eight treatments of the independent variables. I am not sure have Authors used the right experimental design. It is also difficult to assess the interaction of various factors.
We have added the following text in the Introduction:
In line 63-65, the applications of the Taguchi methodology were added as follows:
“The Taguchi methodology has been successfully applied in food, biological and/or biotechnological areas; these include spray drying, fermentations, medicine, pharmacy, food processing, among others [15, 11, 12, 13, 14, 15, 16, 17, 18].”
We amended the Supplementary Materials (S.2. Table 2 and S.3. Figure 1) with relevant information about the Taguchi methodology. A parameter effects plot was then generated from the results of the analysis of means.
The use of small number of experiments is beneficial, but together with the repetitions 24 experiments have been performed! Other methods of planning the experiments reduce their number by using the repetitions only at a focal point.
It is true that there are other experimental designs to reduce the number of experiments, however, the advantages of the Taguchi design over the other methods are the multiple factors that can be evaluated simultaneously and more quantitative information can be extracted from less experimental trials. Taguchi's design is a unique and powerful technique that allows the evaluation of the main effects (without interactions) with a minimum number of experiments providing robust design solutions. So it has been used as an experimental screening design, which are the basis of other experiments. This method can also be applied to designing factorial experiments and analyzing their outcomes.
The Authors used parameters coding, but still this manuscript is difficult to read because they often write all their values. As part of the description of the methodology, it is worth preparing one table with the plan of experiments and coding so as not to repeat it in the figures and tables.
In line 105: we modified Table 1 for better analysis.
Some others comments.
Line 85: “data not shown” - Is the data proprietary? if published, it is a good idea to provide a source.
In line 85: we added relevant application references of the Taguchi methodology
Line 105: The inlet air temperature is given in table 1. What was the temperature of the outlet air in the dryer? This information should also be provided here.
In line 105, the Table 1 shows the experimental design of a Taguchi L8, in which the inlet temperature of the drying process is a variable to consider; however, the outlet temperature corresponds to an effect derived from the drying process [1], so it has been integrated as a variable response in Supplementary Material in Table S.1. in this regard, the following relevant reference was included in the revised manuscript.
[1]. Tolun, A.; Altintas, Z.; Artik, N. Microencapsulation of grape polyphenols using maltodextrin and gum arabic as two alternative coating materials: Development and characterization. J. Biotechnol., 2016, 239, 23-33.
Line 133: What was the examination about, did the mistake creep in?
We corrected the word "drop" for "drew". Additionally, we amended the text as follows:
In line 134-135:
“The water activity is the relative humidity of air (vapor phase water) in equilibrium with the liquid phase water of the sample, in a sealed chamber.”
This test is performed on a device that measures the water activity of a system by equilibrating the liquid phase water in the sample with the vapor phase water in the headspace and measuring the relative humidity of the headspace. The sample is placed in a sample cup which is sealed inside the sample chamber; this device is composed of a fan, a dew point sensor, and an infrared thermometer. The fan helps equilibrate and control the boundary layer conductance of the dew point sensor. The dew point sensor measures the dew point temperature of the air in the chamber, and the infrared thermometer measures the sample temperature. From these measurements, the relative humidity of the headspace is computed as the ratio of the dew point saturation vapor pressure to the saturation vapor pressure at the sample temperature. When the water activity of the sample and the relative humidity of the air are in equilibrium, the measurement of the headspace humidity gives the water activity of the sample. Water activity is the relative humidity of air in equilibrium with a sample in a sealed chamber.
The calibration of the equipment before use was carried out with a standard verification solution with dew point of 0.250 ± 0.003 (13.41 mol / kg LiCl at 25 ° C); we referred to the Aqualab 4TEV equipment user manual, Decagon Devices.
Line 186: Which factor was analysed under "Analysis of variance (ANOVA) using a univariate design was performed to determine the differences between the treatments", if there were 7 factors?
For the univariate analysis, a single factor called “treatment” was used with 8 levels corresponding to each of the combinations of the Taguchi experimental design. This was done to evaluate differences between treatments, subsequently (with the signal-to-noise ratio) the effects of the 7 factors are analyzed (See Supplementary Materials)
We added the following text in “Materials and Methods”:
Line 82-83: “A Taguchi L8 design was performed to obtain the best extraction conditions base upon PC and AOX contents”.
In lines 188-201, previously mentioned in this document.
Line 105: Why are the results of EY and EE presented in table placed in the methodology?
We added the following text in the Table 2 (line 253) and deleted the data of EY and EE in the Table 1 (line 105).
Line 291: The sentence to check
We have edited text in the part “Materials and Methods > X-ray diffraction”
In line 291-292: we modified the following text:
“The X-ray diffraction profiles (XRD) indicate that the MD shows a crystalline structure with nine peaks at 2theta scale = 10°, 11.3°, 13.1°, 14.1°, 17.2°, 21.2°, 26.8°, 28.6° and 30.1° (Figure 3)”
Line 365: Conclusion
We have edited text in the part “Conclusions”
In line 370-374, we changed the text as follows:
“The use of an experimental Taguchi design is a useful methodology that uses a low number of experiments to evaluate the interactions between parameters; this indicates that the maximum TSP content and AOX in the Hibiscus sabdariffa extracts encapsulates were mainly related to the solvent of extraction, extract: carriers agents ratio, and the carriers agents ratio (MD + GA)”
I do not find the Taguchi L8 model used in this manuscript, no mention of it in the discussion of the results, I do not know what model the authors write about, because there is no information about using it to interpret and discuss the results obtained, no model description, no charts, no mathematical analysis, statistical .. It was only used to plan the experiment and what next?
“Supplementary Materials S2 and S3 has been added; this includes information regarding the statistical analysis of the Taguchi design, as well as the discussion of the same therein in the part "Results and Discussion".
Reviewer 2 Report
The manuscript entitled “Use of a Taguchi model in Hibiscus sabdariffa extracts encapsulated by spray-drying” provides information regarding the encapsulation of extracts obtained from different conditions. For that purpose, the authors performed the extraction with water and ethanolic solutions, using distinct temperatures and extraction times. Moreover, the percentage of each carrier (maltodextrin and gum arabic) was varied, as well as the ratio extract/carriers.
In terms of responses, beyond physicochemical properties, the authors also considered the phenolic composition and related antioxidant activity. In this sense, a chromatographic analysis seems indispensable. In fact, the discussion of the results regarding these parameters is directed to the explanation of the possible phenolic composition of the extracts to explain the differences between the antioxidant activity displayed in the different assays. The assessment of the total soluble polyphenols content does not allow specific results for the proposed objectives, because beyond the fact that it does not provide the qualitative analysis of such compounds, this kind of methodology is very susceptible to the interference of other compounds such as sugars.
For these reasons, I would recommend the authors to complete the phenolic compounds analysis in order to valorise the results obtained and make the manuscript suitable for Foods. Otherwise, it would more likely fall within the scope of other journals (related to physicochemical properties of this kind of encapsulates/materials).
Author Response
Reviewer 2
The manuscript entitled “Use of a Taguchi model in Hibiscus sabdariffa extracts encapsulated by spray-drying” provides information regarding the encapsulation of extracts obtained from different conditions. For that purpose, the authors performed the extraction with water and ethanolic solutions, using distinct temperatures and extraction times. Moreover, the percentage of each carrier (maltodextrin and gum arabic) was varied, as well as the ratio extract/carriers.
In terms of responses, beyond physicochemical properties, the authors also considered the phenolic composition and related antioxidant activity. In this sense, a chromatographic analysis seems indispensable. In fact, the discussion of the results regarding these parameters is directed to the explanation of the possible phenolic composition of the extracts to explain the differences between the antioxidant activity displayed in the different assays. The assessment of the total soluble polyphenols content does not allow specific results for the proposed objectives, because beyond the fact that it does not provide the qualitative analysis of such compounds, this kind of methodology is very susceptible to the interference of other compounds such as sugars.
For these reasons, I would recommend the authors to complete the phenolic compounds analysis in order to valorise the results obtained and make the manuscript suitable for Foods. Otherwise, it would more likely fall within the scope of other journals (related to physicochemical properties of this kind of encapsulates/materials).
In line 175, we have added a Figure in the part “Materials and Methods and Results and Discussion” to include chromatographic profiles.
“2.8 Identification of PC by HPLC-DAD
Partial identification of PC was done according to the method proposed using an HPLC Agilent 1260 series system (Agilent Technologies, Santa Clara, CA, USA) equipped with a UV–Vis diode array detector (DAD) [36]. The chromatographic separation was performed in a Poroshell 120 EC-C18 column (4.6mm×150 mm, particle size 2.7 μm) (Agilent Technologies) at a flow rate of 0.4 mL/min using an injection volume of 10 μL. The mobile phases consisted in water containing formic acid (0.1 cm3/100 cm3) as solvent A and acetonitrile as solvent B, following multi-step linear gradient was applied: 0 min, 5% B; 10 min, 23% B; 15 min, 50% B; 20 min, 50% B; 23 min, 100% B; 25 min, 100% B; 27 min, 5% B; 30 min, 5% B; finally, the initial conditions were held for one minutes to equilibrate the system before the subsequent injection. The PCs was monitored in a range of 280-320 nm for the diode array detector”
In line 361: Figure 4. Partial profile of PC analyzed by HPLC-DAD of extracts of Hibiscus (Hibiscus sabdariffa) spray-drying
Round 2
Reviewer 1 Report
The Authors very carefully referred to the comments contained in the review. However, a new improved version of the manuscript was not included.
If the declared changes were included in the new version of this manuscript, the work may be published.
Author Response
The Authors very carefully referred to the comments contained in the review. However, a new improved version of the manuscript was not included.
Thanks for your comment, certainly we included the improved version in the last review, but probably we need to included here too, now is attached too the final version for you approbation, thanks in advance.
The English language and style was checked
If the declared changes were included in the new version of this manuscript, the work may be published.
Reviewer 2 Report
The phenolic compounds chromatographic profile and identification was added to the manuscript. I would just suggest removing “partial identification of polyphenol profile” from the table 3 caption, once the identification is only made in figure 4. In the footnotes of table 3, please add “3” before the respective description “TE, trolox equivalent, dry weight (DW)”.
Author Response
The phenolic compounds chromatographic profile and identification was added to the manuscript.
I would just suggest removing “partial identification of polyphenol profile” from the table 3 caption, once the identification is only made in figure 4. In the footnotes of table 3, please add “3” before the respective description “TE, trolox equivalent, dry weight (DW)”.
We removed the "Partial identification.." in Table 3, and include the 3 in the superscript.
The language and style was reviewed and checked.